# Silane Modification of Mesoporous Materials for the Optimization of Antiviral Drug Adsorption and Release Capabilities in Vaginal Media

**DOI:** 10.3390/pharmaceutics13091416

**Published:** 2021-09-07

**Authors:** Elena Whittle, Araceli Martín-Illana, Raul Cazorla-Luna, Fernando Notario-Perez, María Dolores Veiga-Ochoa, Juan Rubio, Aitana Tamayo

**Affiliations:** 1Faculty of Chemistry, Universidad Complutense de Madrid, Av. Complutense, s/n, 28040 Madrid, Spain; ewhittle@ucm.es; 2Department of Pharmaceutics and Food Technology, Faculty of Pharmacy, Universidad Complutense de Madrid, Plaza Ramón y Cajal s.n, 28007 Madrid, Spain; aracelimartin@ucm.es (A.M.-I.); racazorl@ucm.es (R.C.-L.); fnotar01@ucm.es (F.N.-P.); mdveiga@ucm.es (M.D.V.-O.); 3Institute of Ceramics and Glass, CSIC, Kelsen 5, 28049 Madrid, Spain; jrubio@icv.csic.es

**Keywords:** organic–inorganic hybrid, drug delivery, antiviral, acyclovir

## Abstract

Three different functionalities have been incorporated into mesoporous materials by means of a coupling reaction with the siloxanes 3-glycidoxypropyl-trimethoxysilane (GLYMO), 3-methacryloxypropyl-trimethoxysilane (MEMO), and 3-mercaptopropyl-trimethoxysilane (MPTMS). The disposition of the different functional groups, as well as the interaction mechanism, with the mesoporous substrate has been identified. The amount of the antiviral drug acyclovir (ACV) adsorbed depends not only on the available surface area but also on the chemical or physicochemical interactions between functionalities. The drug adsorption isotherm of the materials functionalized with GLYMO and MPTMS follow mechanisms dependent on the different surface coverage and the possibilities to establish physicochemical interactions between the drug molecule and the functionalities. On the contrary, when functionalizing with MEMO, the dominant adsorption mechanism is characteristic of chemically bonded adsorbates. The ACV release kinetics is best fitted to the Weibull model in all the functionalized materials. When the MTPMS is used as a functionalizing agent, the drug diffusion occurs at low kinetics and homogeneously along the mesoporous channels.

## 1. Introduction

Mesoporous materials are highly versatile in many different technological areas such as catalysis [1], gas sensors [2], as adsorbents for wastewater management [3], and in energy production and storage applications [4]. Research in the pharmaceutical and biomedical applications of mesoporous materials has experienced an outstanding increase during recent years, with those based on silica or alumina being the most traditionally used. The mesoporous SBA-15 and MCM-41 silica have been widely exploited [5,6], although some other bioactive glasses and ceramics [7] or titanium oxide [8] present excellent characteristics to gain leverage. Among the mesoporous organic materials, it withstands the polypropylene polymers, polystyrene, poly(α, β-l malic acid), poly(ethylene glycol)-block-poly (β-caprolactone), and ethyl cellulose [9,10]. As most of these materials are biocompatible, the applications in biomedicine are spread over different fields such as magnetic resonance imaging (MRI) contrasting agents [11] and nanomedicine for the treatment of different diseases [12,13,14,15]. Organic–inorganic hybrids synthesized by sol–gel offer the possibility to obtain materials with high surface area, selected pore size, and tunable hydrophilic–hydrophobic characteristics, as well as the possibility of bulk or surface modification whilst combining the inherent characteristics of the organic and inorganic matrixes [16,17,18,19]. In this sense, recent trends in mesoporous materials development such as drug delivery systems (DDS) have been directed from pure inorganic structures [15,20,21] through hybrid materials [22] toward metal–organic frameworks [23] and porous organic frameworks [24].

Current investigations have been focused on the synthesis of new nanomaterials with mesoporous structures of a large pore diameter with specific surface functionalities to achieve a stimuli response in smart pharmaceutical formulations [25]. Surface functionalization with silane coupling agents is an effective strategy to fill the available surface with the desired functionality. It is also widely used to enhance the adsorption and desorption capabilities of the materials, also allowing the modulation of the hydrophilic or hydrophobic properties of the substrate [6,26]. The use of coupling agents in the local delivery of anti-HIV-drugs has been reported in several works, either to increase the solubility of the support [27,28,29], to achieve a pH-dependent release by modulation of the affinity of the drug for the support [13,14], or to provide a molecular recognition function to target the HIV reservoirs [30]. Vaginal administration is a well-recognized strategy for the local drug delivery of anti-HIV drugs due to the high permeability of the vaginal mucosa and also because this route overpasses the first-pass hepatic metabolism suffered by the drugs administered by the oral route. For vaginal drug delivery, surface functionalization is now focused on promoting the penetration of the drug support through the mucus layer (generally carried out by means of polyethylene glycol grafting [31]) and to prevent protein adsorption and subsequent undesired bioaccumulation [32].

Silica-based mesoporous hybrid materials overcome some of the main drawbacks related to the incompatibility of the mesoporous silicas in some biological environments. Hybridization via surface modification with organic molecules has been proved to be a feasible strategy to reduce the high density of the silanol groups on the surface of the silica particles, thus modifying its cytotoxic effect. Apart from providing a specific functionality, the conjugation of silica particles with polyethylene glycol to formulate an hybrid material inhibits hemolysis [33], and the functionalization with various silane molecules allows minimization of the agglomeration and deposition under physiological conditions [34].

As a functionalization agent, the alkoxysilane 3-(glycydoxypropyl)trimethoxy silane (GLYMO), which contains three methoxy groups on one side and an epoxy ring (oxirane) at the end of the aliphatic branch, is widely used as a grafting agent to enhance the anchoring sites in silica-derived materials [35,36]. GLYMO has become very popular as a coupling or grafting ligand because of its facile nucleophilic ring-opening reaction of the pendant oxirane functional group, thus allowing the binding to molecules with different functionalities. It provides a multipoint covalent attachment either with amino, thiol, carboxylic acids, or even phenolic groups [37]. Similarly, the hydrophilic/hydrophobic characteristics of SiO_2_ [38], TiO_2_ [39], metal nanoparticles [40,41], and multiwall carbon nanotubes (MWCNTs) [42] can be easily tuned through silane grafting with 3-(methacryloyloxypropyl)trimethoxysilane (MEMO). Thiol functionalization, on the other hand, has widely exploded in molecular recognition [43] and biosensing [44]. In the material’s surface functionalized with the silane 3-(mercaptopropyl)-trimethoxysilane (MPTMS), the so-called thiol-ene click chemistry involving the coupling of a –SH termination with a C=C double bond, a large number of biomacromolecules such as carbohydrates [45], enzymes [46], or aptamers [47] have been successfully immobilized. In this work, these three siloxane coupling agents have been incorporated onto the surface of a mesoporous hybrid material with a bimodal pore size distribution, and their effect on the adsorption mechanism and pharmacokinetics of the antiviral drug acyclovir has been determined.

## 2. Materials and Methods

### 2.1. Synthesis of the Functionalized Mesoporous Hybrid Materials. Characterization Methods

The bimodal mesoporous materials were synthesized following the procedure described by Martin-Illana et al. [14]. In brief, the silicon alkoxyde triethoxysilane (ABCR, Karlsruhe, Germany) and the silicone polymer polydimethylsiloxane (ABCR, Karlsruhe, Germany) were used as raw materials in a weight ratio of 70:30. Isopropanol (iPrOH, Sigma Aldrich, St. Louis, MO, USA, 98%) was used as the solvent. The silicon alkoxyde and the polymer were mixed in a reactor thermostated at 70 °C with the solvent iPrOH, H_2_O, and hydrochloric acid (HCl, Fluka, Charlotte, NC, USA, 32%) all in a molar ratio of 4.5:3:0.05:1 (iPrOH:H_2_O:HCl:triethoxysilane). After 1 h at 70 °C under constant stirring at 300 rpm, the sol was poured into a propylene container and, after 30 min, a solution of NH_4_OH 1 M was slowly added (caution, vapor gases emerged). The gels were first dried at 25 °C, 50 °C, and 120 °C until constant weight. Then, the dried gels were pyrolyzed under a constant N_2_ flow of 180 mL·min^−1^ at 700 °C with 2 h of dwelling time to obtain the bimodal mesoporous materials. The mesoporous materials were then crushed and sieved below 50 μm. The coupling agents 3-glycidoxypropyl-trimethoxysilane (GLYMO), 3-mercaptopropyl-trimethoxysilane (MPTMS), and 3-methacryloxypropyl-trimethoxysilane (MEMO) were used. GLYMO and MPTMS were hydrolyzed for 1 h in H_2_O at 25 °C. MEMO was hydrolyzed in ethanol (EtOH) at pH 1 for 6 h. Then, 1 g of the hybrid particles was added to the solutions containing 1, 1.5, 3, 5, and 10% silane w/w. After 30 min under vigorous stirring, the surface-modified particles were filtered and then dried at 50 and 120 °C to remove the adsorbed H_2_O and promote chemisorption. These mesoporous particles demonstrated noncytotoxic effects even at high concentrations [14]. For sample labeling, the percentage amount of silane added to the grafting solution was used, followed by a G for the materials functionalized with GLYMO, by an M for the materials functionalized with MEMO, and by an S for the materials functionalized with the MPTMS coupling agent. The microstructure of the mesoporous particles was observed by using a field emission scanning electron microscope (FE-SEM, Hitachi S4700, Tokyo, Japan) operating at 20 kV.

The characterization of the mesoporous materials was carried out by means of FTIR spectroscopy performed in a Perkin-Elmer model BX spectrophotometer (Perkin-Elmer, Waltham, MA, USA) equipped with a MIRacle^TM^ accessory for attenuated total reflectance (ATR) measurements, obtaining the spectra as the average of 32 collections with a spectral resolution of 2 cm^−1^. The thermal decomposition of the functionalized mesoporous materials was studied through coupled thermogravimetry and differential thermal analysis in a TA Q600 thermobalance (TA Instruments, New Castle, DE USA) fed with a constant air flow of 100 m·min^−1^. The differential thermal analysis allows the evaluation of the adsorption mechanism (chemisorption or physisorption) of the silane coupling agents. Then, 100 mg of the material was placed into alumina crucibles and the mass change and heat flow difference were continuously recorded while heating at 10 °C/min to 700 °C, which is the maximum temperature used for obtaining the mesoporous supports. The specific surface areas (SSA) were calculated by the Brunauer–Emmett–Teller (BET) equation [48] from the nitrogen adsorption isotherms obtained at 77 K in a Tristar 3000 (Micromeritics, Norcross, GA, USA). Pore size distributions (PSDs) were obtained by the application of the BJH procedure (Barrett, Joyner, and Halenda) [49] to the desorption branch of the isotherm.

### 2.2. Drug Loading and Release Studies

The antiviral drug acyclovir (ACV) was then loaded into the dried material. Then, 0.2 g of the mesoporous material was suspended in a solution of a certain concentration of ACV (5 mg of ACV/100 mL of H_2_O). At different time intervals, the solutions were filtered and the spectra of the solutions were collected in a UV-Vis spectrometer (Perkin Elmer Lambda 40, Waltham, MA, USA). The absorbance maximum at 254 nm was used to determine the amount of ACV loaded at each time. Once the time for maximum ACV absorption was determined, 100 mg of particles was then immersed in 25 mL of Simulated Vaginal Fluid (SVF) [50] thermostated at 37 °C under steady stirring. Then, 500 μL aliquots of the SVF solution were taken at certain time intervals and replaced with 500 μL of fresh SVF. As in the drug loading experiments, the amount of ACV in solution was determined by UV-vis spectroscopy by using a new ACV calibration obtained by using SVF as the solvent.

In Figure 1, the synthesis procedure is schematically represented, where all the reagents involved in the preparation of the mesoporous hybrid are detailed, the functionalization with the specified silanes, and the drug loading and release experiments.

## 3. Results

The microstructure of the mesoporous material, as observed by electron microscopy, is shown in Figure 2. In Figure 2a, a low-resolution image of the obtained particles is shown, where it can be appreciated that they consist of small particulate aggregates with voids that conform to the mesoporous structure. At high resolution (Figure 2b), the two types of pores can be observed. Notice that the rough surface of the particles, as observed in the high-resolution images, may be due to the gold sputtering. In Figure 2c, a representative image of the functionalized material is shown, in this case, corresponding to the one functionalized with 3% GLYMO. Similar images are obtained in the remaining cases. There, it is observed that the morphology of the particles before and after functionalization remains the same, thus excluding the possibility of the contribution to the microstructure of small aggregates of polycondensed silane. Full details of the microstructure of the bare mesoporous material can be found in Ref. [14].

### 3.1. Infrared Spectroscopy

The mesoporous materials functionalized with the coupling agents containing different functional groups were characterized by FTIR spectroscopy in the ATR mode. All the spectra of the functionalized mesoporous materials are summarized in Appendix A. In Figure 3, the FTIR spectra obtained in these materials are plotted in the spectral ranges where the most significant bands of the functional groups of the silane molecules appeared. In the case of the materials functionalized with GLYMO, the two most significant regions have been represented, the region between 1800 and 1500 cm^−1^ (Figure 3a) and the low-frequency region (1300–850 cm^−1^, notice the axis break) in Figure 3b. In all the materials, independent of the functionalization, it is possible to detect the presence of C=C bonds at about 1640 cm^−1^ and attributed to the partial decomposition of the polymeric structure of the mesoporous hybrid (Figure 3).

In the mesoporous materials functionalized with the GLYMO silane (Figure 3b), the attention is focused on the bands appearing at 908 cm^−1^ and attributed to the epoxy antisymmetric deformation [51], as well as the stretching vibration of the C–O bond in opened rings at 1270 cm^−1^ [52]. In these spectra shown in Figure 3a,b, the oxidation of the opened epoxide ring is also observable in the form of C=O bonds as characteristic bands appearing at 1720 cm^−1^ and 1250 cm^−1^ [53]. From these different spectra (surface-functionalized mesoporous hybrids–bare mesoporous substrate), band deconvolution has been carried out to find the relative intensities of these spectral bands in order to evaluate the relative number of the abovementioned functional groups present on the surface. The fitting procedure has been validated through χ^2^ minimization. In Figure 4a, the ratio of the open to closed epoxy rings calculated from the ratio of the intensity of the bands located at 1270 and 908 cm^−1^ is presented. There, it is observed that after a first increase in the relative number of the opened rings, there is a minimum ratio (more closed rings than opened rings) at intermediate GLYMO concentrations. From the intensity of the band appearing at 1250 cm^−1^ (Figure 4a, right axis), and corresponding to the oxidized epoxy groups, it is also possible to deduce that, precisely at these concentrations, the amount of GLYMO incorporated is maximum just considering that all the groups will oxidize equally.

The characteristic vibrations of MEMO are shown in the spectra at 1635 cm^−1^ (–C=C– group), 1703 cm^−1^ (vibration of the –C(=O)–O– group bonded to silanol), and another mode at 1723 cm^−1^ assigned to the stretching vibrations of C=O in the conjugated structure [54,55,56]. In Figure 4b (left axis), the intensity ratio of the carbonyl group in the acrylate structure is shown forming condensed and conjugated bonds with respect to those groups associated with hydroxyl functionalities. A general increase in the conjugated structures is observed with a maximum at 3% silane. The number of –C=C– groups (Figure 4b, right axis) is also an indicative of the concentration of MEMO incorporated on the surface of the mesoporous hybrids.

It is difficult to quantitatively determine the concentration of –SH on the surface of the mesoporous hybrids due to the weak peak intensity of –SH in FTIR. SH-stretching of mercaptosilane appears as a very weak band at 2570 cm^−1^ [57] and is more noticeable in the material containing the highest amount of the silane. The peak appears split into two because of the intermolecular hydrogen bindings between adjacent S-H groups shifting the natural frequency to lower wavelengths [57]. As shown in Figure 4c, the maximum free S-H groups with respect to the hydrogen-bonded thiol groups occurs at a silane concentration of 3% and then decreases gradually.

### 3.2. Thermal Analysis

The differential thermal analyses (DTAs) carried out in the oxidant atmosphere (Figure 5) show that in the case of the functionalization with GLYMO and MEMO, the oxidation of the functionalizing agent occurs at lower temperature as the concentration of silane increases, indicating that at the lowest concentrations, the silane is bonded directly to the surface of the mesoporous material, whereas by increasing the concentration of the silane, a multilayered functionalization takes place. The opposite behavior is observed in the case of the MTPMS, where a slight increase in the degradation temperature is observed when the amount of incorporated silane is beyond 1.5%.

From the thermogravimetric curves, the effective amount of coupling agent (in mmol) with respect to the theoretical percentage incorporated during the synthesis was calculated and is plotted in Figure 6. Although the thermogravimetric analysis provides information about the amount of silane (in weight) incorporated onto the surface of the hybrid materials, the normalization to a molar amount was executed. There, it can be observed that both the MEMO and MPTMS are incorporated in similar amounts, being larger than the incorporated amount of GLYMO

In Figure 6b, it is observed that the combustion enthalpy calculated from the DTA curves during the thermal treatment in the oxidant atmosphere increases continuously with the amount of silane and follows a similar trend as the amount of silane incorporated in the case of the GLYMO and MEMO silanes. On the contrary, the functionalization of the mesoporous material with MPTMS leads to a maximum in the oxidation enthalpy at 1.5% MPTMS, indicating different adsorption mechanisms depending on the amount of the coupling agent incorporated. 

### 3.3. Nitrogen Adsorption/Desorption

The nitrogen adsorption desorption isotherm of the bare hybrid material was published elsewhere [14] and demonstrated the mesoporous characteristic of the material, with a bimodal shape of the mesoporous distribution, with pores ranging between 4 and 35 nm. The bimodality of the pore size distribution is characterized by two peaks in the curves, showing a gap between peaks that reflects the filling by the adsorbate of the low- sized and high-sized mesopores. After functionalization with the silanes, the specific surface area (collected in Table 1) varies not only with the amount of silane incorporated but with the functional group of the silane molecule as well.

As it is observed in Table 1, the mesoporous hybrids functionalized with the silane coupling agent GLYMO experience a slight increase in the SSA with respect to the bare material at silane concentrations up to 1.5%. This increase in the SSA is attributed to the formation of small aggregates of the silane molecule.

As shown in Figure 7, all the materials remain mesoporous at all the functionalization percentages, although the pore size is slightly reduced. The hybrid materials functionalized with GLYMO show a uniform PSD with similar pore size and volumes at all the concentrations. When MEMO is used as a functionalization agent, it is observed that, at the minimal MEMO content, the pore size distribution is broader at low pore sizes and then becomes narrower, and the total adsorbed volume decreases with the amount of silane. The functionalization of the mesoporous hybrids with MPTMS leads to a progressive reduction in the mean pore size as the amount of silane increases up to a minimum mean pore size of 3.6 nm, which is found for the material containing 1.5% MPTMS. Afterward, the PSD becomes broader and the adsorbed pore volume decreases as well.

### 3.4. Drug Loading and Release

The amount of ACV adsorbed on the surface of the pores of the mesoporous materials has been studied by collecting the UV spectrum of a solution of ACV at different intervals. From the absorbance of the solution, the concentration of the ACV adsorbed was then calculated. The uptake of ACV per gram of mesoporous hybrid was calculated according to the following equation:(1)$$C_{eq}=\frac{C_{0}-C_{e}}{m\;}\;V$$
where *C_eq_* is the equilibrium adsorption capacity (mg·g^−1^), *C*_0_ and *C_e_* are, respectively, the initial concentration and the equilibrium concentration (mg·L^−1^) of ACV in solution, *V* is the volume of solution (L), and *m* is the dry weight of mesoporous material (g). The calculated *C_eq_* is plotted in Figure 8, where the dotted line corresponds to the bare material. In Figure 8a–c, a different adsorption mechanism and total amount of ACV adsorbed depending upon the number of functionalization and functional groups can also be observed. In the materials functionalized with GLYMO, after a fast uptake of ACV, and especially in these materials containing 1.5 and 3% GLYMO, a plateau in the absorption is observed, a fact that does not occur in these materials with MEMO and MPTMS where a progressive increase in ACV with time characterized the absorption isotherm.

Taking into account that the materials possess a different available surface area, in Figure 8d,e, the amount of ACV absorbed per m^2^ of mesoporous material has been plotted. There, it is appreciated that the minimal differences in the amount of ACV adsorbed per available area are found in the materials with GLYMO as a coupling agent, and also, the absorption capability is lowest among the remaining functionalities. On the contrary, the maximum absorption ability is found in the materials whose functionalization is provided by the silane MPTMS.

The absorption of the drug molecules over the surface of the mesoporous materials involves the transport of the drug from the aqueous phase to the surface of the material, the diffusion of the drug toward the pores, and the physical, physical-chemical, or chemical adsorption of the drug on the active sites existing on the pore surfaces. According to the adsorbed amount of ACV, and represented in Figure 8, the most plausible adsorption mechanism has been discerned by fitting the curves to different adsorption kinetic models: Lagergren’s pseudo-first-order [58,59], pseudo-second-order [60], and the Elovich kinetic models [61].

Lagergren’s equation is the earliest known one describing the adsorption rate based on the adsorption, and it has been widely used for the adsorption of an adsorbate (the drug molecules) from an aqueous solution. This model is expressed as:(2)$$\ln\left(\frac{C_{\infty }}{C_{\infty }-C_{eq}}\right)=k_{lag}t$$
where the terms *C_∞_* and *k_lag_* are the maximum absorption capability and the rate constant of the first-order adsorption (1/min), respectively. If Equation (2) is fulfilled, the sorption process is controlled by the diffusion through the liquid film surrounding the mesoporous material.

The pseudo-second-order model is based on the assumption that the rate of occupation of the active sites is proportional to the square of the number of unoccupied sites. The model is widely used when a chemical bonding exists between adsorbate molecules and active surface sites on the pores, which controls the adsorption [62]:(3)$$C_{eq}=\frac{C_{\infty }{}^{2}k_{2s}t}{1+C_{\infty }t}$$
where *k*_2*s*_ is the pseudo-second-order rate constant of adsorption. A different kinetic equation of chemisorption, known as the Elovich equation, assumes a variation in the energetics of chemisorption with the extent of surface coverage [63]. It was established by Zeldowisch [61] and represented by:(4)Ceq=1bln(a b)+1bln(t)
where *a* is related to the initial adsorption rate and *b* is related to the surface coverage. 

According to the fitting values obtained by the application of these models and collected in Appendix A, it can be concluded that depending on the degree of functionalization and the type of coupling agent, the adsorption of the ACV over the surface of the mesoporous materials occurs following different mechanisms. To discern which mechanism fits better, we have used both the r^2^ criteria and the Fischer statistics test, which describes the overall significance of the regression model (F). The F value is the ratio of the mean regression sum of squares divided by the mean error sum of squares and varies from zero to an arbitrarily large number.

The best fits and the kinetic constants are summarized in Table 2. The nonfunctionalized material can be fitted to the Elovich model, as well as the material functionalized with the minimum amount of the silane GLYMO. By increasing the amount of GLYMO, the best fits occur by the application of the Lagergren and pseudo-second-order models, and then at the maximum functionalization degree, the Elovich equation applies again. This transference to a different absorption mechanism also occurs in the materials functionalized with MTPMS, whereas in the case of the mesoporous materials containing the coupling agent MEMO, the pseudo-second-order model, i.e., the assumption of a chemical bonding mechanism, is the only one that applies.

Once the materials were loaded with the maximum amount of drug, the release profiles in SVF were followed by UV-Vis spectroscopy (Figure 9). The maximum percentage of ACV released in the first 120 min in contact with the SVF was obtained in the mesoporous materials functionalized with 1 and 5% GLYMO. In the case of the materials containing the methacryloxy functionality, the maximum release was observed at 3% functionalization and the profile was similar to the nonfunctionalized sample. Those materials functionalized with MPTMS presented their maximum release capability in the first 2 h in the material containing 1.5% silane.

As it was carried out with the drug absorption profiles, the mechanism of drug release was determined by fitting the curves shown in Figure 9 to the most common mathematical drug release models [64]. The *first-order* kinetic model is usually applied to porous matrices. In this model, the release rate is only dependent on the concentration of the drug in the releasing media, and it is expressed as:(5)$$\frac{C_{t}}{C_{\infty }}=1-e^{-K_{1st}t}$$
where *C_t_* and *C_∞_* are the cumulative absolute amount of drug released at time t and at infinite time, respectively. In the Korsmeyer–Peppas kinetic model, the drug release is described as a function of time, where the *n* exponent in Equation (6) indicates the mechanism responsible for the drug release:(6)$$\frac{C_{t}}{C_{\infty }}=K_{KP}t^{n}$$

When the value of *n* is less than or equal to 0.5, the drug diffusion through the SVF dominates the release. For n values between 0.5 and 1.0, an “anomalous transport phenomenon” occurs, which is based on diffusion and the structural modification of the dosage form (*n* = 0.5 corresponds to Higuchi model). A values of *n* equal to 1.0 (“transport case II”) or over 1.0 (“transport super-case II”) characterizes drug release profiles that are due only to structural changes in the formulation. The Weibull kinetic model is an empirical equation that can be applied to almost all dissolution/release processes of drug systems:(7)$$\frac{C_{t}}{C_{\infty }}=1-\frac{e^{-{\left(t-T_{1}\right)}^{b}}}{a}$$
where the parameter *T_1_* is the lag time (usually zero), *a* is a scale parameter describing the time dependence, and *b* takes different values as a function of the shape of the dissolution curves (*b* = 1 indicates exponential curve, *b* = 2 fits for sigmoidal curves, and *b* = 3 indicates parabolic curves) [65].

The kinetics data obtained through the fitting to the abovementioned kinetic models are collected in Appendix A. Except for the nonfunctionalized material, whose best fit is obtained through the application of the first-order model, the remaining materials are better-adjusted to the Weibull kinetic model. The data obtained from the best fits are also summarized in Table 3. Values of *b* in the range of 0.75–1.0 indicate a combined mechanism that is frequently encountered in release studies, and it is characteristic of Fickian diffusion. An increase in *b* reflects a decrease in the disorder in the medium [66].

In the Weibull model, when *b* = 1, the shape of the curve is an exponential profile and the kinetic constant may be expressed as *k* = 1/*a*. The evolution of the kinetic parameter *k* (1/*a*) with the amount of silane is plotted in Figure 10a. There, a slight acceleration in the drug release can be appreciated at intermediate concentrations of the silanes GLYMO and MEMO, whereas, in the case of the functionalization with the MPTMS silane, at these intermediate concentrations, the drug release occurs more slowly.

The b parameter, which is directly related to the diffusion mechanism of the drug throughout the SVF might also be assimilated to a kinetic constant, as shown in Figure 10b. In the materials functionalized with GLYMO and MEMO, the low b values observed at the intermediate silane concentrations indicate that the diffusion occurs in a fractal substrate, whereas, in the case of MPTMS, an increase in this value is observed, indicating the diffusion in the ordered or Euclidean space [66]. The high *b* values found in the fitting of the nonfunctionalized material indicate Fickian or first-order diffusion, thus confirming the model that fitted best according to the statistical analysis.

This section may be divided by subheadings. It should provide a concise and precise description of the experimental results, their interpretation, as well as the experimental conclusions that can be drawn.

## 4. Discussion

From the results shown in the previous section, it is clear that the different functionalities provided by the coupling agents confer to the mesoporous materials a specific behavior against the drug loading and release capabilities. The silane GLYMO is exceptional among other silanes in the sense that the oxirane ring is highly reactive. As in other silanes, the coupling mechanism is assumed to occur between the silanol groups on the surface of the mesoporous material and the hydrolyzed silanol groups of GLYMO. However, a reaction between the epoxide ring and the silanol groups is also possible. As demonstrated in the convolution analysis of the FTIR spectra (Figure 4a), at intermediate GLYMO concentrations (3 and 5%), there are more closed rings than open rings, thus increasing the possibility of reaction between the oxirane groups and the silanol groups on the surface of the mesoporous material (notice that if the oxirane ring is open before the coupling reaction, this reaction cannot occur). In addition, after the silane has been grafted, there may also be some interactions involving the epoxide group and the surface that interferes with ring opening. The combustion enthalpy (Figure 6b) calculated from the thermal analysis shown in Figure 5 and the amount of coupling agent (Figure 6a) reveals that none of the mechanisms in the degradation of the silane molecule are different, thus excluding the possibility of additional interactions between the oxirane ring and the surface of the mesoporous material.

The available functional groups in the GLYMO-containing material are therefore either opened or closed oxirane rings and, among the opened oxiranes, there might also be some oxidized C=O groups appearing especially at concentrations higher than 3% GLYMO. This variation in the functional groups provided by the GLYMO molecule might be responsible for the different drug loading mechanisms observed depending on the amount of silane incorporated. At low and high silane concentrations (1% and 10%), the Elovich model [63], which takes into account the variations in the energetics and the surface coverage, fits better. The increased number of functional groups leads to the evolution of a mechanism controlled by diffusion of the drug toward the inner pores to a model that considers chemisorption of the drug molecules [62], a situation that occurs in the material functionalized with 5% GLYMO, which possess the maximum number of closed oxirane rings, thus suggesting the occurrence of an *N*-alkylation reaction between these epoxide rings and the purine base of the ACV molecule [67]. In Figure 11a, the FTIR of the mesoporous material functionalized with 3% GLYMO is shown, as well as the same material but after being loaded with ACV. The appearance of new bands at 1420 cm^−1^ and 1270 cm^−1^ corroborates the formation of new bonds between the ACV molecule and the functionalized substrate via *N*-alkylation reactions. In the same figure, the FTIR spectra of ACV in the same spectral range are also plotted, where it can be confirmed that these new bands are exclusively due to new bonds and not because of the presence of the ACV in the material.

In the case of the functionalization with the MEMO silane, the analysis of the FTIR (Figure 4b) shows that, in addition to the general increase in the intensity of the –C=C– bonds in the methacrylate group, there is an increased number of silane molecules whose functional group is interacting with the silanol groups located at the surface of the mesoporous material, i.e., they are not available to interact with the drug molecule. This configuration of the silane molecule on the surface of the mesoporous substrate favors the coupling of the silane in a multilayered disposition, as reported by Hoang et al. in the functionalization of silica nanoparticles with the MEMO silane [68]. The formation of hydrogen bonds between the carbonyl group in the methacrylate with the silanol groups of the surface of the mesoporous material is also evidenced by the decrease in the temperature at which the oxidation of the silane takes place, as reflected in Figure 5b. The possible occurrence of any polycondensation reactions between the silanol groups of the trialkoxysilane and the surface of the material would result in an increase in the oxidation temperature, as demonstrated by several authors [69]. 

The MEMO molecules are then adsorbed on the surface of the material occupying the pores of the smallest diameter (Figure 7b). The adsorption of the ACV fits better to the pseudo-second-order reaction, suggesting that a chemical interaction might occur between the methacrylate groups of the MEMO silane and the ACV molecules via the formation of new amide groups, similar to that which occurred in hydroxyethylmethacrylate hydrogels loaded with ACV [70]. In Figure 11b, the FTIR spectrum of the mesoporous material containing 3% MEMO after being loaded with the ACV is shown, where the new bands appearing at 1630 cm^−1^ and 1270 cm^−1^ verify the interaction between the functional groups of the MEMO silane and the drug molecule. The adsorption kinetic constant increases gradually with the amount of silane incorporated up to 5% silane that drops drastically, a fact that is attributed to the decreased SSA. Despite this increase in the kinetic constant, the *C_∞_* value (Table 2), which is the maximum concentration of adsorbed ACV, decreases with the amount of MEMO incorporated, a result that is attributed to the increased hydrophobicity of the material with the functionalization [71]. 

Thiol functionalization evolves differently than in the remaining silanes. The differential thermal analysis (Figure 5c) shows almost no difference in the temperature at which the decomposition of the silane takes place (except for the material containing 1% MPTMS), and the combustion enthalpy is quite similar as well (Figure 6b). As shown in Figure 4c, there occurs a maximum number of free –SH groups and hydrogen-bond-interacting –SH at 3% silane, similar to that in the GLYMO functionalization, but the combustion enthalpy reveals a different coupling mechanism only at silane concentrations beyond 1%. The PSDs shown in Figure 7c are not as homogenous as in the remaining materials, showing a preferred adsorption in the smaller pores at low MPTMS concentrations, whereas by increasing the MPTMS amount above 3%, the PSDs become broader. 

Sananes–Israel et al. [72] demonstrated that the nature of the functional group of the coupling agent strongly affects the morphology and structure of the grafted layers. In the case of MPTMS, only at 60 °C, a perfect monolayer is obtained, whereas at low temperature, a partial filling of the monolayer was observed [72]. At low silane concentrations (1 and 1.5%), the adsorption of the ACV follows an adsorption model that depends on the surface coverage, with a great difference in the *a* parameter (related to the kinetic constant) between the two samples. The adsorption of ACV occurs generally by donor–acceptor interactions [26], and the MPTMS is known for increasing the acidic characteristic of the surface of the silica [71]. Increasing the amount of MPTMS then favors the amount of ACV adsorbed per m^2^, as evidenced in Figure 8a–c.

With regard to the pharmacokinetics data, all the functionalized materials present a different releasing mechanism than the nonfunctionalized one. The slight acceleration in the drug release at intermediate concentrations found in the materials functionalized with GLYMO and MEMO suggests that the respective equilibrium of the *N*-alkylation reaction and amidation reactions is shifted to the nonalkylated and nonamidated forms because of the change in the releasing media (pH = 4.2). The absence of any chemical interaction (just donor–acceptor interactions) between the MPTMS and ACV leads to a release mechanism dominated by the diffusion through a quasi-ordered surface (the drug molecule is released from an homogeneous medium from the chemical point of view, with no functional groups altering the charge balance). All of these results suggest that, for a controlled release of the drug loaded in materials capable of establishing chemical equilibriums between the available functional groups and the drug molecule, it is necessary to control the kinetics of this equilibrium to achieve a sustained release. However, when acid–base interactions are established, as has occurred in the case of MPTMS functionalization, the *b* values ranging between 0.6 and 0.75 in the Weibull kinetic model indicate that the release of the drug occurs at the lowest kinetics and the diffusion takes place homogeneously throughout the releasing medium, where any other interaction (either repelling or attracting groups) that could modify the diffusion pathways does not exist [66]. MPTMS functionalization is thus considered the proper candidate for the development of smart vaginal films for antiviral drug delivery.

## 5. Conclusions

The use of coupling agents to modify the surface of mesoporous hybrid materials is proved to be an easy strategy to incorporate different functionalities to a substrate that is designed to act as a carrier of drug molecules. Increasing amounts of coupling agents are not always translated to a larger number of functional groups. With the GLYMO coupling agent, which is one of the most popular for incorporating a highly reactive oxirane group, the increased number of closed oxirane cycles increased the possibility of interacting with the silanol groups, either on the surface of the mesoporous hybrid or the hydrolyzed functional groups of the silane molecule, thus minimizing the number of functionalities available to interact with the drug molecule. These closed rings are capable of forming *N*-alkylation reactions with the ACV purine base and the functional group of the GLYMO molecule. On the other hand, the incorporation of the functionality provided by the MEMO coupling agent leads to an increase in the hydrophobic characteristic of the substrate because of the interaction of the polar groups in the methacryloxy functionality with the silanol groups encountered on the surface of the material. Nevertheless, it is possible to establish a chemical interaction via new amide groups with the nitrogenated base of ACV. MPTMS, on the other hand, results as the most promising silane molecule to achieve a smart response in vaginal media. MPTMS increases the acidic characteristic (electron acceptor) of the mesoporous substrate but possess the inconvenience of the difficulty of achieving a perfect monolayer adsorption. On the contrary, the drug release mechanism is dominated by the diffusion throughout the channels and, despite no chemical bonding existing between the drug molecule and the coupling agent, just a physicochemical interaction, the release occurs at lower kinetics.

## Figures and Tables

**Figure 1 pharmaceutics-13-01416-f001:**
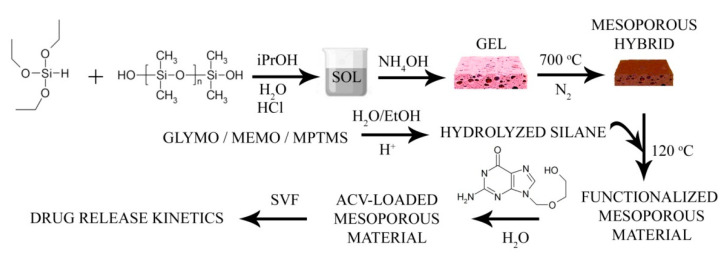
Schematic representation of the synthesis of the mesoporous material, functionalization with the silane molecules, and the drug loading and release procedures.

**Figure 2 pharmaceutics-13-01416-f002:**
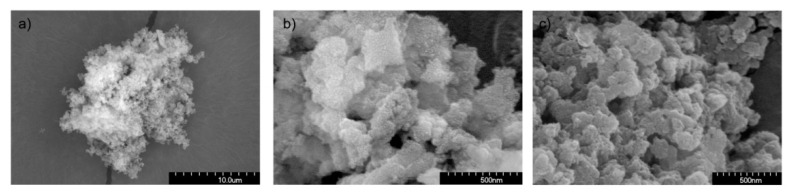
Scanning electron microscope images of (**a**,**b**) mesoporous material (low and high resolution, respectively) and (**c**) the functionalized material containing 3% GLYMO.

**Figure 3 pharmaceutics-13-01416-f003:**
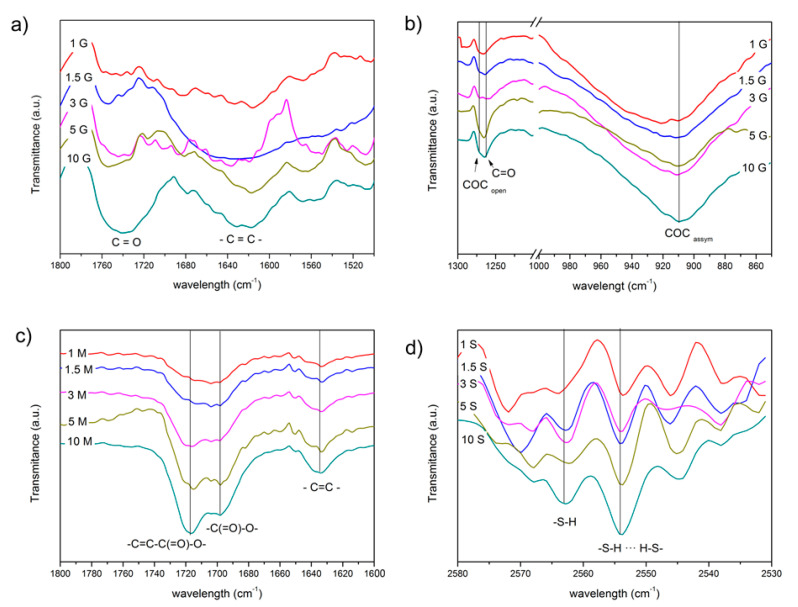
Detailed regions of the FTIR spectrum of the mesoporous materials functionalized with (**a**,**b**) GLYMO, (**c**) MEMO, and (**d**) MPTMS silane at different concentrations.

**Figure 4 pharmaceutics-13-01416-f004:**
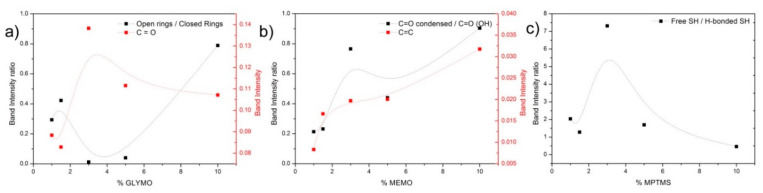
(**a**) Relative intensity ratio of the bands centered at 1270 to 908 cm^−1^ (left axis) and relative intensity of the band at about 1250 cm^−1^ (right axis); (**b**) relative intensity ratio of the bands centered at 1723 to 1703 cm^−1^ (left axis) and relative intensity of the band at about 1635 cm^−1^ (right axis); (**c**) relative intensity ratios of the bands centered at 2570 to 2550 cm^−1^. Lines are plotted to guide the eye.

**Figure 5 pharmaceutics-13-01416-f005:**
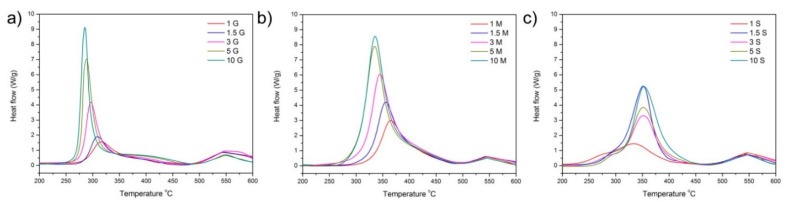
Differential thermal analysis curves of the materials functionalized with (**a**) GLYMO, (**b**) MEMO, and (**c**) MPTMS at different concentrations (indicated in the labeling as % weight).

**Figure 6 pharmaceutics-13-01416-f006:**
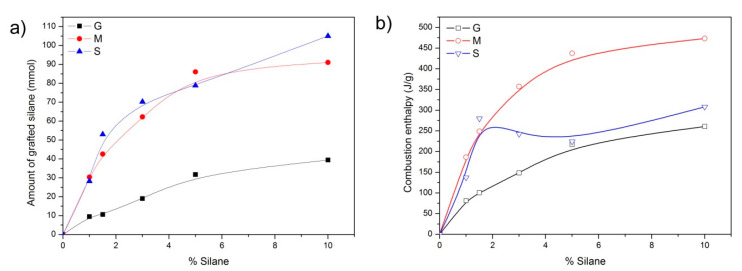
(**a**) Amount of coupling agent (in mmol) at each concentration of silane incorporated during the synthesis, and (**b**) combustion enthalpy of the silanes. Lines are plotted to guide the eye.

**Figure 7 pharmaceutics-13-01416-f007:**
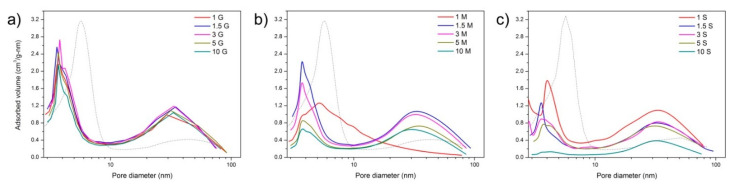
BJH pore distribution of the mesoporous hybrids functionalized with different percentage amounts of (**a**) GLYMO, (**b**) MEMO, and (**c**) MPTMS silane molecules at different concentrations (indicated in the labeling as % weight). Dash-dotted lines represent the BJH distribution of the unfunctionalized mesoporous hybrid particles.

**Figure 8 pharmaceutics-13-01416-f008:**
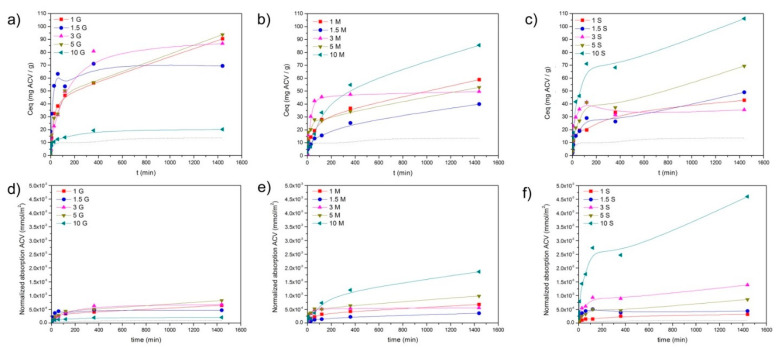
*Ceq* (in mg/g) of the mesoporous materials functionalized with (**a**) GLYMO, (**b**) MEMO, and (**c**) MPTMS. Dotted line corresponds to the nonfunctionalized mesoporous material. Normalized absorption of ACV (in mmol/m^2^) with respect to the surface area of the materials functionalized with (**d**) GLYMO, (**e**) MEMO, and (**f**) MPTMS.

**Figure 9 pharmaceutics-13-01416-f009:**
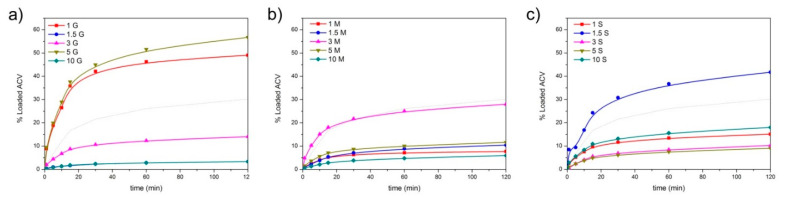
Release profiles of ACV in SVF functionalized with (**a**) GLYMO, (**b**) MEMO, and (**c**) MPTMS. Dotted line corresponds to the nonfunctionalized mesoporous material.

**Figure 10 pharmaceutics-13-01416-f010:**
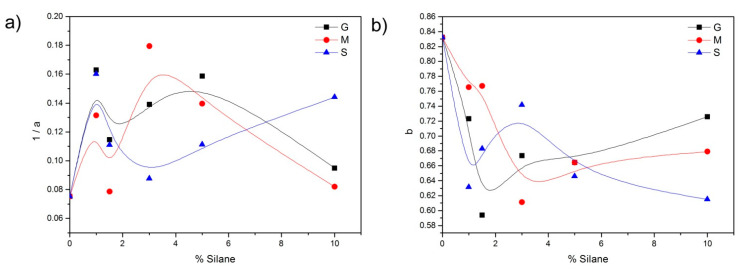
(**a**) Weibull kinetic parameter *k* (1/a) as a function of the silane concentration, and (**b**) Weibull transport parameter b as a function of the silane concentration.

**Figure 11 pharmaceutics-13-01416-f011:**
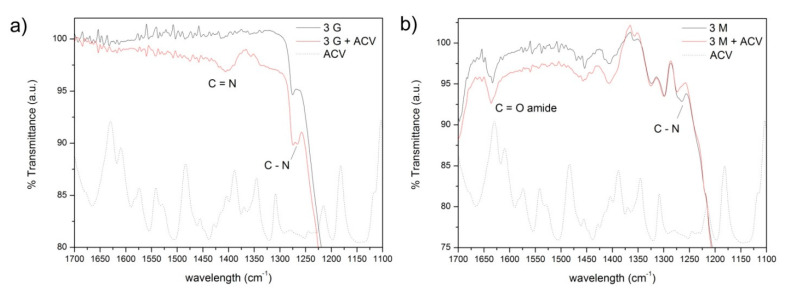
FTIR spectra of the mesoporous materials loaded with ACV and functionalized with (**a**) GLYMO and (**b**) MEMO. Dotted lines show the FTIR spectra of the drug molecule ACV.

**Table 1 pharmaceutics-13-01416-t001:** Specific surface area of the functionalized mesoporous hybrids.

Concentration Silane (wt %)	G		M		S	
	SSA (m^2^/g)	Pore Volume (cm^3^/g)	SSA (m^2^/g)	Pore Volume (cm^3^/g)	SSA (m^2^/g)	Pore Volume (cm^3^/g)
0	610	1.13	610	1.13	610	1.13
1	618	1.20	387	0.87	595	1.22
1.5	657	1.23	499	1.15	360	0.82
3	572	1.14	397	1.01	339	0.74
5	502	1.10	239	0.67	254	0.69
10	434	0.98	204	0.63	67	0.26

**Table 2 pharmaceutics-13-01416-t002:** Fitting constants and model applied to the absorption curves of the mesoporous materials.

Material	Model			
Nonfunctionalized	Elovich	a	b	r^2^
		4.54	0.59	0.957
1 G		7.61	0.09	0.937
	Lagergren	k_lag_	c_∞_	r^2^
1.5 G		6.261 × 10^−2^	64.6	0.887
3 G		7.910 × 10^−3^	85.9	0.953
	Pseudo-second-order	k_2s_	c_∞_	r^2^
5 G		1.011 × 10^−4^	96.1	0.952
	Elovich	a	b	r^2^
10 G		14.71	0.45	0.963
	Pseudo-second-order	k_2s_	c_∞_	r^2^
1 M		1.373 × 10^−4^	58.9	0.914
1.5 M		1.504 × 10^−4^	41.7	0.932
3 M		9.591 × 10^−4^	51.5	0.963
5 M		3.130 × 10^−3^	37.0	0.600
10 M		3.650 × 10^−5^	100.6	0.977
	Elovich	a	b	r^2^
1 S		19.23	0.25	0.847
1.5 S		337.12	0.29	0.795
	Pseudo-second-order	k_2s_	c_∞_	r^2^
3 S		2.234 × 10^−4^	62.3	0.896
5 S		3.160 × 10^−4^	44.0	0.896
10 S		2.051 × 10^−4^	96.4	0.930

**Table 3 pharmaceutics-13-01416-t003:** Fitting constants and model applied to the drug release curves of the mesoporous materials.

Material	Model				
Nonfunctionalized	First Order	C_∞_ (mg)	k_1st_ (min^−1^)		r^2^
		0.98	5201 × 10^−2^		0.990
	Weibull	C_∞_	b	a	r^2^
1 G		1.86	0.72	6.14	0.992
1.5 G		0.19	0.59	8.73	0.997
3 G		0.69	0.67	7.20	0.995
5 G		2.78	0.66	6.30	0.995
10 G		0.17	0.73	10.54	0.995
1 M		0.32	0.77	7.61	0.995
1.5 M		0.46	0.77	12.71	0.996
3 M		1.21	0.61	5.57	0.998
5 M		0.50	0.66	7.16	0.995
10 M		0.28	0.68	12.19	0.997
1 S		0.84	0.63	6.24	0.995
1.5 S		2.39	0.68	9.01	0.971
3 S		0.57	0.74	11.41	0.991
5 S		0.53	0.65	8.99	0.995
10 S		1.04	0.61	6.94	0.995

## Data Availability

Raw data are available upon request to the corresponding author.

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
