# Peer review of "Silane Modification of Mesoporous Materials for the Optimization of Antiviral Drug Adsorption and Release Capabilities in Vaginal Media"

_pharmaceutics, 2021, doi:10.3390/pharmaceutics13091416_

Round 1

Reviewer 1 Report

The manuscript “Silane modification of mesoporous materials for the optimization of antiviral drug adsorption and releasing capabilities in vaginal media” prepared by Whittle et al. reports fabrication and comprehensive characterization of mesoporous materials functionalized with three different agents for acyclovir releasing into simulated vaginal fluid.

The paper is well-written and logically built in terms of methodology. This will be of specific interest of those involved in optimization of existing drug delivery porous particles and design of new ones.

Nevertheless the submitted materials should be subjected a minor revision before being accepted for publication in the Pharmaceutics journal.

Please see the remarks below.

  1. The introduction section is recommended to be enriched with speculations on biocompatibility/biodegradability of the carrier used comparing to other porous media applied in drug delivery. Is there a need for these materials to possess not only cytotoxity absence but also to be taken by living tissues as their own parts? If not please explain why.
  2. Please clarify what the “bimodal” term means while referring to mesoporous particles.
  3. The paper is fruitful for scholars looking for description of the methodology of the mesoporous particles synthesis and their functionalization with different polymer molecules as well as loading drugs in their pores. However, a novelty aspect is not prominent enough. In particular, the FTIR spectroscopy has been widely utilized to characterize the functionalized porous materials. This is recommended to compare the FTIR spectra with those collected by Raman spectroscopy, which is attractive complementary method of molecular analysis.
  4. All the Figures are required to be improved in quality, e.g. inscriptions must be increased.
  5. Please double-check English grammar, e.g. missed prepositions and wrong tenses.

Author Response

The authors acknowledge the reviewer the effort to improve the quality of our manuscript by pointing out several aspects that were not sufficiently clear in our previous version. Below, you can find the responses to the questions raised by the reviewer as well as some details on the precise changes made in the revised manuscript:

Reviewer 1: The introduction section is recommended to be enriched with speculations on biocompatibility/biodegradability of the carrier used comparing to other porous media applied in drug delivery. Is there a need for these materials to possess not only cytotoxity absence but also to be taken by living tissues as their own parts? If not please explain why.

Authors: According to the reviewer suggestion, we have enriched our introductory part with some references regarding the absence of cytotoxicity of the hybrid particles. Hybridization has been proved to be a strategy to reduce the cytotoxicity of the silica a well as the surface functionalization, as we referred in the revised introductory section. For vaginal drug delivery, however, the thick and semipermeable cervicovaginal mucus limits the access access of pathogens to the upper genital tract and squamous epithelium. Although some of the current investigations are focused on enhanced penetration throughout the cervicovaginal mucus,t he absence of cytotoxicity of the functionalized particles was demonstrated in our previous investigations (Martin-Illana, et al. Amino Functionalized Micro-Mesoporous Hybrid Particles for the Sustained Release of the Antiretroviral Drug Tenofovir. Materials 2020, 13, 3494.).

Below, you can find some of the changes made in the introduction part to address the cytotoxicity and biocompatibility issues of the mesoporous particles for vaginal drug delivery:

Current investigations are focused on the synthesis of new nanomaterials with mesoporous structures of a large pore diameter with specific surface functionalities to achieve a stimuli response in smart pharmaceutical formulations [25]. Surface functionalization with silane coupling agents is an effective strategy to fill the available surface with the desired functionality. It is also widely used to enhance the adsorption and desorption capabilities of the materials also allowing the modulation of the hydrophilic or hydrophobic properties of the substrate [6,26]. The use of coupling agents in local delivery of anti HIV-drugs has been reported in several works, either for increasing the solubility of the support [27-29], to achieve a pH dependent release by modulation of the affinity of the drug for the support  [13,14] or to provide molecular recognition function to target the HIV reservoirs [30]. Vaginal administration is a well-recognized strategy for local drug delivery of anti-HIV drugs due to the high permeability of the vaginal mucosa and also because this route overpasses the first-pass hepatic metabolism suffered by the drugs administered by the oral route. For vaginal drug delivery, surface functionalization is now focused promoting the penetration of the drug support through the mucus layer (generally carried out by means of polyethylene glycol grafting [31]) and to prevent protein adsorption and subsequent undesired bioaccumulation [32]

Silica-based mesoporous hybrid materials overcome some of the main drawbacks related to the incompatibility of the mesoporous silicas in some biological environments. Hybridization via surface modification with organic molecules has been proved to be a feasible strategy to reduce the high density of the silanol groups on the surface of the silica particles thus modifying its cytotoxic effect. Apart from providing a specific functionality, conjugation of silica particles with polyethylene glycol to formulate an hybrid material inhibits hemolysis [33] and the functionalization with various silane molecules allows minimizing agglomeration and deposition under physiological conditions [34].”

Reviewer: Please clarify what the “bimodal” term means while referring to mesoporous particles.

Authors: In the revised version of the manuscript, we have clarified the term “bimodal pore size distribution” when referring to our mesoporous materials. The following sentence has been included to clarify this term:

Authors: The bimodality of the pore size distribution is characterized by two peaks in the curves showing a gap between peaks that reflects the filling by the adsorbate of the low sized and high sized mesopores

Reviewer: The paper is fruitful for scholars looking for description of the methodology of the mesoporous particles synthesis and their functionalization with different polymer molecules as well as loading drugs in their pores. However, a novelty aspect is not prominent enough. In particular, the FTIR spectroscopy has been widely utilized to characterize the functionalized porous materials. This is recommended to compare the FTIR spectra with those collected by Raman spectroscopy, which is attractive complementary method of molecular analysis.

Authors: The authors completely agree with the reviewer in the sense that Raman spectroscopy is highly attractive and provides substantial information which can complement the one extracted from the FTIR spectra. However, one of the main problems with Raman spectroscopy is that in some samples, the fluorescence background is quite high and the fluorescence signal overlaps the bands corresponding to the chemical bonds present in the materials. This is the case of the mesoporous substrates analyzed and characterized in this work. We have tried to obtain high quality spectra by irradiating the samples with an Ar-ion laser operating at 514 nm and with a 785 nm direct Diode laser. In all the cases we failed to obtain high quality spectrum that deserve to be published. Below, the authors include some of the Raman spectra of the mesoporous functionalized samples, in all the cases with 3% of siloxane coupling agent.

From these spectra, the authors consider quite bold trying to extract any conclusion out of it and it could lead to misinterpretation of the result.

A possible solution which could minimize the strong fluorescence background could be to shift the laser wavelength to a lower energetic one. In future, we will look for access to this facility since, as we pointed out before, and in complete agreement with the reviewer, it should be studied and compared with other spectroscopic techniques.

Reviewer: All the Figures are required to be improved in quality, e.g. inscriptions must be increased.

Authors: The figures have been uploaded again with an improved quality

Reviewer: Please double-check English grammar, e.g. missed prepositions and wrong tenses.

Authors: An extensive revision of the English grammar has been carried out all throughout the manuscript and the detected errors have been corrected.

Reviewer 2 Report

The reviewed manuscript titled ‘Silane modification of mesoporous materials for the optimization of antiviral drug adsorption and releasing capabilities in vaginal media’ (pharmaceutics-1339862) presents the method of the synthesis of hybrid mesoporous materials functionalized with three different coupling agents. The manuscript is well written, the used methods are adequate, but the discussion of obtained results is questionable. From the description of the synthesis method of the discussed materials and then from results section I’m not sure how Authors think the modifications took place. Reading the manuscript firstly I thought that the functionalization was in the course of chemical reactions, then Authors wrote about the adsorption of coupling agents (e.g. line 419). Please write clearly about the synthesis and add the scheme of a synthetic route of hybrid materials. Next, Authors also didn’t wrote about the form/morphology of obtained materials. The manuscript should be complemented by the discussion of their morphology. Moreover, the names of coupling agents are incorrectly written, the abbreviations are also confused (e.g. GLUMO).

Line 150: Authors wrote that the oxidation of opened epoxide to carbonyl group took place, giving as an evidence the presence of band at 1250 cm-1 in the FTIR spectrum. It is not the most characteristic absorption band for C=O. Could you provide the FTIR spectra of GLYMO functionalized materials in the rang about 1700 cm-1 to see if really the carbonyl is present in the structure of materials. How could Authors explain the possible oxidation reaction in this non-oxidative conditions? Moreover, in Supplementary Materials, there is putted Fig S1 presenting FTIR spectra, but the figure title is incomplete.

Line 413: in the MEMO structure there is no carboxyl groups!

Line 171 and 481: Authors wrote acrylate group whereas they did not use any acrylate compounds!

Could you provide any evidence (e.g. FTIR spectrum) confirmed that the reaction between methacrylate group of MEMO functionalized material and amine group present in ACV took place? It is rather impossible to transfer ester group to amide group (especially in such described conditions).

Author Response

The authors acknowledge the reviewer the effort to improve the quality of our manuscript by pointing out several aspects that were not sufficiently clear in our previous version. Below, you can find the responses to the questions raised by the reviewer as well as some details on the precise changes made in the revised manuscript:

Reviewer: The reviewed manuscript titled ‘Silane modification of mesoporous materials for the optimization of antiviral drug adsorption and releasing capabilities in vaginal media’ (pharmaceutics-1339862) presents the method of the synthesis of hybrid mesoporous materials functionalized with three different coupling agents. The manuscript is well written, the used methods are adequate, but the discussion of obtained results is questionable. From the description of the synthesis method of the discussed materials and then from results section I’m not sure how Authors think the modifications took place. Reading the manuscript firstly I thought that the functionalization was in the course of chemical reactions, then Authors wrote about the adsorption of coupling agents (e.g. line 419). Please write clearly about the synthesis and add the scheme of a synthetic route of hybrid materials.

Authors: Accordingly to the reviewer suggestion, we have included a new figure (Figure 1 in the revised version of the manuscript) where it is schematically represented the experimental procedure. In the experimental section, it is also specified that the drying at 120ºC was carried out to remove the adsorbed H2O on the surface of the mesoporous substrate and promote the chemisorption reactions between the silanol groups present in the surface of the mesoporous materials and the hydrolyzed silane molecules. In addition, it is specified that the differential thermal analysis has been used as a tool to evaluate the adsorption mechanism of the silane coupling agents.

Reviewer: Next, Authors also didn’t wrote about the form/morphology of obtained materials. The manuscript should be complemented by the discussion of their morphology.

Authors: As it was stated in the experimental section, the materials were synthesized following the procedure described by Martin-Illana et al. (ref 14: Martin-Illana, et a. Amino Functionalized Micro-Mesoporous Hybrid Particles for the Sustained Release of the Antiretroviral Drug Tenofovir. Materials 2020, 13, 3494).There, a comprehensive description of the microstructure is given, a fat that has been specified, in the revised version of the manuscript. Moreover, according to the reviewer suggestion, in the result section of the revised version, the authors included the SEM images of the obtained material, before and after functionalization with the silane molecules.

Reviewer: Moreover, the names of coupling agents are incorrectly written, the abbreviations are also confused (e.g. GLUMO).

Authors: The authors thanks the reviewer for noticing these unintentional errors that were unadvertised by the authors. The whole manuscript has been subjected again to an Englisgh revision in terms of grammar and language to correct these mistakes.

Reviewer: Line 150: Authors wrote that the oxidation of opened epoxide to carbonyl group took place, giving as an evidence the presence of band at 1250 cm-1 in the FTIR spectrum. It is not the most characteristic absorption band for C=O. Could you provide the FTIR spectra of GLYMO functionalized materials in the rang about 1700 cm-1 to see if really the carbonyl is present in the structure of materials. How could Authors explain the possible oxidation reaction in this non-oxidative conditions? Moreover, in Supplementary Materials, there is putted Fig S1 presenting FTIR spectra, but the figure title is incomplete.

Authors: The authors have included the FTIR spectra in the region where the C=O band is shown. The plot of this spectral region now takes part of Figure 3.

Reviewer: Line 413: in the MEMO structure there is no carboxyl groups!

Authors: Thanks for noticing the mistake. It was actually a carbonyl group. This has been corrected in the revised version of the manuscript

Reviewer: Line 171 and 481: Authors wrote acrylate group whereas they did not use any acrylate compounds!

Authors: The authors agree with the reviewer in the sense that probably the sentence was not well constructed. Accordingly, we have clarified that we referred to carbonyl groups contained in the acrylate structure of the silane.

Reviewer: Could you provide any evidence (e.g. FTIR spectrum) confirmed that the reaction between methacrylate group of MEMO functionalized material and amine group present in ACV took place? It is rather impossible to transfer ester group to amide group (especially in such described conditions).

Authors: As the reviewer recommended, we included a new figure in the revised manuscript (FTIR) where the occurrence of the new bands because of the formation of new bonds between the functionalities of the mesoporous material and the ACV molecules. This new figure is Figure 11 in the revised version.

Reviewer 3 Report

In this manuscript, the authors have modified mesoporous materials through coupling reactions with 3 different chemicals. They investigated how the drug loading and release properties were affected by the different modification approaches. They found the drug adsorption mechanism were different between the modified mesoporous materials. The drug was absorbed on the GLYMO and MPTMS modified surface through physiochemical interactions while it was absorbed on the MEMO modified materials mainly based on chemically bonded adsorbates. The publication of this manuscript might be of some value for other researchers in this field. However, there are some minor issues in the manuscript which need to be addressed before publication.

  1. In Figure 1, the curves should be labeled more clearly to avoid any confusion. For example, based on the legend, the numbers “1, 1.5, 3, 5, 10” represent the concentration of the chemicals. However, what is the unit? Is it mol/L, mg/mL or w/w%? For clear comparison of the transmittance of the different curves, the authors had better provide a vertical line to mark the wavelength of the characteristic groups.
  2. In Figure 2, the points seem highly scattered. What is the confidence value for the curve fitting with the five scattered points? Are there any statistical results, such as R2, to support the accuracy of the fitting? How many repeat samples have the authors tested? What’s the standard deviation for each data point?
  3. In Figure 3, the authors had better change the label 1G to 1% GLYMO for better readability.
  4. In Figure 4, the authors had better change the label G to GLYMO for better readability. How many parallel samples have the authors tested? What’s the standard deviation for each data point?
  5. In Figure 5, the authors had better change the label 1G to 1% GLYMO for better readability.
  6. In Figure 6, the authors had better add the label and units for x axis for picture a), b), and c); and add the label and units for y axis for picture b), c), e), and f). The sizes of the picture a), b), c) should be adjusted to match the sizes of the pictures d), e), and f). How many parallel samples have the authors tested? What’s the standard deviation for each data point?
  7. In Figure 7, it is hard to tell the dotted line from solid lines, which needs to be addressed. Moreover, it is hard to see the “dotted line” in picture b). How many parallel samples have the authors tested? What’s the standard deviation for each data point? Have the authors made the comparison with any statistical methods, and what is the confidence for the comparison.
  8. In Figure 8, the points seem highly scattered. What is the confidence value for the curve fitting with the five scattered points? Are there any statistical results, such as R2, to support the accuracy of the fitting? How many repeat samples have the authors tested? What’s the standard deviation for each data point?

Author Response

The authors acknowledge the reviewer the effort to improve the quality of our manuscript by pointing out several aspects that were not sufficiently clear in our previous version. Below, you can find the responses to the questions raised by the reviewer as well as some details on the precise changes made in the revised manuscript:

Reviewer: In Figure 1, the curves should be labeled more clearly to avoid any confusion. For example, based on the legend, the numbers “1, 1.5, 3, 5, 10” represent the concentration of the chemicals. However, what is the unit? Is it mol/L, mg/mL or w/w%? For clear comparison of the transmittance of the different curves, the authors had better provide a vertical line to mark the wavelength of the characteristic groups.

Authors: The referred figure (now Figure 3 in the revised version of the manuscript) has been modified. It has been included the right labeling of the samples (1G, 1.5, 3G, 5G, 10 G and so on, accordingly to the sample labeling and as specified in the text) and also, as the reviewer suggested, a straight line has been included in the plots to highlight the position of the functional groups.

Reviewer: In Figure 2, the points seem highly scattered. What is the confidence value for the curve fitting with the five scattered points? Are there any statistical results, such as R2, to support the accuracy of the fitting? How many repeat samples have the authors tested? What’s the standard deviation for each data point?

Authors: The deconvolution procedure has been performed by using the minimum squared method in a single FTIR spectrum for each sample. The chi square criterion has been used to evaluate the goodness of the fitting (it has been also indicated in the text). In the revised version of the manuscript, it is specified that the lines are just plotted to guide the ey3

Reviewer: In Figure 3, the authors had better change the label 1G to 1% GLYMO for better readability.

Authors: The authors have followed the same labeling criteria for all the figures. It is indicated the sample labeling, which correspond to we weight percent silane added in each case. In the figure legend, it has been however indicated

Reviewer: In Figure 4, the authors had better change the label G to GLYMO for better readability. How many parallel samples have the authors tested? What’s the standard deviation for each data point?

Authors: The enthalpy values, as recorded in Figure 6 in the revised version (previously it was Figure 4), were calculated from the integration of the DSC data. Just one sample was recorded in each case so no statistical values were obtained

Reviewer: In Figure 5, the authors had better change the label 1G to 1% GLYMO for better readability.

Authors: As previously commented, the authors have followed the same labeling criteria. Nevertheless, in the figure legend it has been specified that in each case, it refers to the weight percent silane

Reviewer: In Figure 6, the authors had better add the label and units for x axis for picture a), b), and c); and add the label and units for y axis for picture b), c), e), and f). The sizes of the picture a), b), c) should be adjusted to match the sizes of the pictures d), e), and f). How many parallel samples have the authors tested? What’s the standard deviation for each data point?

Authors: The precise changes in the figure have been carried out. Drug loading kinetic was performed in triplicate and the data point was taken as the averaged UV-Vis spectrum.

Reviewer: In Figure 7, it is hard to tell the dotted line from solid lines, which needs to be addressed. Moreover, it is hard to see the “dotted line” in picture b). How many parallel samples have the authors tested? What’s the standard deviation for each data point? Have the authors made the comparison with any statistical methods, and what is the confidence for the comparison.

Authors: As it was performed in the case of drug loading, the kinetics of the drug release was carried out in triplicate and the data points were taken as the averaged UV-Vis spectrum. Unfortunately, no statistical treatment was carried out.

Reviewer: In Figure 8, the points seem highly scattered. What is the confidence value for the curve fitting with the five scattered points? Are there any statistical results, such as R2, to support the accuracy of the fitting? How many repeat samples have the authors tested? What’s the standard deviation for each data point?

Authors: Data of Figure 8 (now Figure 10 in the revised version) correspond to the values obtained from the fittings to the kinetic models. The r2 and Chi squared values obtained in each fitting are collected in Table 3

Round 2

Reviewer 3 Report

The authors have answered most of my questions and concerns. Now the manuscript is in good state for publication.